

# Integrated omics analysis reveals the immunologic characteristics of cystic Peyer's patches in the cecum of Bactrian camels

Xiao shan Wang, Pei xuan Li, Bao shan Wang, Wang dong Zhang and Wen hui Wang

College of Veterinary Medicine, Gansu Agricultural University, Lanzhou, Gansu, China

Corresponding author
Wen hui Wang, www777@163.com

## ABSTRACT

*Bactrian camels* have specific mucosa-associated lymphoid tissue (MALT) throughout the large intestine, with species-unique cystic Peyer's patches (PPS) as the main type of tissue. However, detailed information about the molecular characteristics of PPS remains unclear. This study applied a transcriptomic analysis, untargeted metabolomics, and 16S rDNA sequencing to compare the significant differences between PPS and the adjacent normal intestine tissues (NPPS) during the healthy stage of three young *Bactrian camels*. The results showed that samples from PPS could be easily differentiated from the NPPS samples based on gene expression profile, metabolites, and microbial composition, separately indicated using dimension reduction methods. A total of 7,568 up-regulated and 1,266 down-regulated differentially expressed genes (DEGs) were detected, and an enrichment analysis found 994 DEGs that participated in immune-related functions, and a co-occurance network analysis identified nine hub genes (*BTK, P2RX7, Pax5, DSG1, PTPN2, DOCK11, TBX21, IL10,* and *HLA-DOB*) during multiple immunologic processes. Further, PPS and NPPS both had a similar pattern of most compounds among all profiles of metabolites, and only 113 differentially expressed metabolites (DEMs) were identified, with 101 of these being down-regulated. Deoxycholic acid (DCA; VIP = 37.96, log2FC = −2.97, *P* = 0), cholic acid (CA; VIP = 13.10, log2FC = −2.10, *P* = 0.01), and lithocholic acid (LCA; VIP = 12.94, log2FC = −1.63, *P* = 0.01) were the highest contributors to the significant dissimilarities between groups. PPS had significantly lower species richness (Chao1), while Firmicutes (35.92% ± 19.39%), Bacteroidetes (31.73% ± 6.24%), and Proteobacteria (13.96% ± 16.21%) were the main phyla across all samples. The LEfSe analysis showed that *Lysinibacillus, Rikenellaceae_RC9_gut_group, Candidatus_Stoquefichus, Mailhella, Alistipes,* and *Ruminococcaceae_UCG_005* were biomarkers of the NPPS group, while *Escherichia_Shigella, Synergistes, Pyramidobacter, Odoribacter, Methanobrevibacter, Cloacibacillus, Fusobacterium,* and *Parabacteroides* were significantly higher in the PPS group. In the Procrustes analysis, the transcriptome changes between groups showed no significant correlations with metabolites or microbial communities, whereas the alteration of metabolites significantly correlated with the alteration of the microbial community. In the co-occurrence network, seven DEMs (M403T65-neg, M329T119-neg, M309T38-neg, M277T42-2-neg, M473T27-neg, M747T38-1-pos, and M482t187-pos) and 14 genera (*e.g., Akkermansia, Candidatus-Stoquefichus, Caproiciproducens,* and *Erysipelatoclostridium*) clustered much more tightly, suggesting dense interactions.

The results of this study provide new insights into the understanding of the immune microenvironment of the cystic PPS in the cecum of *Bactrian camels*.

## INTRODUCTION

Camels are highly adapted to the desert ecosystem of extreme temperatures, scarce vegetation, and limited food and water resources, and have a relatively high resistance to a wide range of pathogens (*Hussen & Schuberth, 2021*). In China, the two-humped *Bactrian camel* (*Camelus bactrianus*), belonging to the subfamily Camelinae, is mainly distributed in arid or semiarid areas like the Gobi desert (*Khomeiri & Yam, 2015*), where camel milk and meat are important to the local economy. *Bactrian camels* have evolved a special gastrointestinal system for resource conservation and a unique mucosal immune system formation to fight against harsh environmental conditions (*ZhaXi et al., 2014*). It is well known that the tight regulation of intestinal immune responses is pivotal for individual functional maintenance (*Peterson & Artis, 2014*). Peyer's patches (PPS) are specialized mucosa-associated lymphoid tissue (MALT), forming a central part of the inductive site of the mucosal immune system; PPS take and trap gut macromolecules and microorganisms (*Jung, Hugot & Barreau, 2010*). Therefore, the characterization of mucosal immune mechanisms is important to an understanding of disease pathogenesis and the development of effective vaccines against gastrointestinal infections. However, a comprehensive understanding of the PPS in the two-humped camel immune system is still unexplored.

A previous study reported four distinct types of PPS in *Bactrian camels*: nodular, faviform, cup-shaped, and cystic, with the nodular and cystic PPS types unique to *Bactrian camels* (*Qi et al., 2011*). Previously, we reported the structure and distribution of the MALT throughout the large intestine of *Bactrian camels* using anatomical and histological methods and observed that the shape of PPS gradually changed from ''scrotiform'' to ''faviform'' along the large intestine with the unique scrotiform, or cystic, as the primary PPS type observed in the ileocecal orifice (*ZhaXi et al., 2014*). The cecum has been suggested as the leading inductive site for mucosal immune responses in the large intestine of the *Bactrian camel* (*Wang et al., 2022*). Previous studies have applied the whole transcriptome to research regional PPS heterogeneity in rats and pigs and both studies reported the distinctive gene expression patterns of PPS in different intestinal locations (*Maroilley et al., 2018*; *Phillips et al., 2021*). Using transcriptomic sequencing, researchers have also revealed the common regulatory patterns of all follicular cells isolated from PPS and immunized peripheral lymph nodes, and also found that the up-regulation or down-regulation of genes was attenuated substantially between the different groups (*Georgiev et al., 2018*). Nevertheless, there is still an incomplete understanding of the basic gene expression patterns of PPS in the cecum of *Bactrian camels* during a steady state.

Symbiotic flora densely colonizes the mammalian gastrointestinal mucosal surface, which stimulates the host's immune responses. A large and diverse number of microorganisms, including commensal bacteria and pathogenic bacteria, have been found in the gastrointestinal tract (*Lin et al., 2021*), for example, *Lactobacillus*, a probiotic species that enhances the barrier function of the intestinal epithelium (*Livingston et al., 2010*), and *Salmonella*, a kind of pathogenic taxa that diminishes the tight junctions of the epithelium, leading to bacterial invasion (*Zhang et al., 2014*). Generally, gut microbiota effects host growth, infection prevention, immune regulation, metabolism, and intestinal health. Previous studies have mainly focused on evaluating the impact of bacterial flora on animal development and nutrition. For instance, *He et al. (2019b)* found that the diversity and stability of gut microbial communities increased with the age of *Bactrian camels*, and the community compositions differed by age. Recently, more research has been devoted to revealing the internal interactions between gut taxa and intestinal mucosa. By comparing commensal microbial composition differences between the aggregated lymphoid nodule area (ALNA) and the ileal PPS, we previously observed that the host's intestinal microenvironment is selective for the symbiotic bacteria colonizing the corresponding sites, which could impact the physiological functions of the site (*Zhang et al., 2020*). However, it is still unknown if there are associations between colonized flora and the gut-expressed genes in PPS.

Since the symbiotic interaction between gut microbiota and the host is reflected in specific metabolic signatures, metabolomics approaches are expected to help unveil the gut microbial-host co-metabolites (*Nicholson et al., 2012*). A previous study found that microbial metabolism provides the nutrients needed for the everyday activities of the host, including vitamins and short-chain fatty acids (*Rowland et al., 2018*). A separate study found that changes to the co-metabolism of bile acids, choline, and purines has been associated with host obesity (*Palau-Rodriguez et al., 2015*). The intestinal microbiota can metabolize some drugs into specific metabolites and alter the systematic bioavailability of other drugs (*Zhang, Zhang & Wang, 2018*). The composition and variation of metabolites in PPS are not yet understood.

This study uses integrated approaches to deeply research the specific immunologic environment of PPS in the cecum of *Bactrian camels*. Three clinically healthy *Bactrian camels* were randomly selected from large camel groups; PPS and NPPS samples from the cecum were collected, separately. Transcriptomic sequencing, non-targeted metabolomics, and 16S barcoding sequencing strategies were used to analyze gene expression patterns, the composition of metabolites, and the diversity of the symbiotic microbial community. A correlation analysis of differentially expressed genes, metabolites, and bacterial taxa provides new insights into the immunologic characteristics of PPS in *Bactrian camels*.

## MATERIALS & METHODS

### Ethics approval

The Animal Care and Use Committee (IACUC) of the College of Veterinary Medicine of Gansu Agricultural University approved all experimental procedures (Approval No: GSAU-AEW-2020-0010). All efforts were made to minimize animal suffering.

## Sample collection

Three clinically healthy *Bactrian camels* (4–5 years old, two females and one male) were randomly selected among the camel group. All the camel specimens were brought from the slaughterhouse in Minqin County, Gansu province; the animals did not suffer starvation before being anesthetized with sodium pentobarbital (20 mg/kg) and exsanguinated. After opening the abdominal cavity, the ileocecal orifice and cecum were separated from the large intestine. The samples collected for subsequent transcriptomic sequencing were gently washed with sterilized saline to remove the residue, except for the samples collected for the detection of microbiota and metabolites. Under a sterile operation, five cystic Peyer's patches (PPS) and five adjacent normal intestine tissues (NPPS) from the cecum of each camel were randomly selected before the capsule contents of PPS and the mucus of the NPPS were collected. The PPS contents from a single camel were mixed as a single biological sample; the mucus of NPPS was mixed and treated as a single control. Finally, three biological PPS samples (PPS-1, PPS-2, and PPS-3) and three NPPS samples (NPPS-1, NPPS-2, and NPPS-3) were collected and put into the 2.5 ml cryopreserved tubes, which were immediately sent to Gene Denovo Biotechnology Co., Ltd. (Guangzhou, China) in dry ice for all the subsequent experiments.

## Transcriptomic sequencing and analyzing

Total RNA was extracted from each tissue using the Trizol reagent kit (Invitrogen, Carlsbad, CA, USA) according to the manufacturer's protocol. Agilent 2100 Bioanalyzer (Agilent Technologies, Palo Alto, CA, USA). RNA quality, quantity, and integrity were determined using RNase-free agarose gel electrophoresis. Then, mRNA was enriched with synthetic oligo (dT) and fragmented using a fragmentation buffer. The cDNA was generated by reverse transcription from enriched mRNA by random primers, and DNA polymerase I, RNase H, dNTP, and buffer were used to synthesize the second-strand cDNA. The QiaQuick PCR extraction kit (Qiagen, Venlo, Netherlands) was used for cDNA purification, followed by end repair, adding poly (A), and sequencing adapter ligation. The final products were size-selected by agarose gel electrophoresis and sequenced using the Illumina HiSeq2500.

The raw data were first filtered by Fastp v0.18.0. Reads containing adapters, reads containing more than 10% of unknown nucleotides (N), and reads containing more than 50% low quality ($Q$-value $\leq$ 20) bases were all removed. Then, the remaining short, high-qualified reads were aligned to the reference genome of *Bactrian camels* (https://www.ncbi.nlm.nih.gov/genome/10741?genome_assembly_id=212083) using HISAT v2.2.4 with default parameters. The mapped reads of each sample were assembled using StringTie v1.3.1 in a reference-based approach and the fragments per kilobase of transcript per million mapped reads (FPKM) for all assembled transcripts were calculated by RSEM. DESeq2 was applied to analyze the differential expressed genes (DEGs) between two groups. The gene was considered significant between groups when the false discovery rate (FDR) was lower than 0.5 and the absolute fold change (|FC|) was higher than 1. The EnrichR was used to conduct GO (http://www.geneontology.org/) and KEGG (http://www.kegg.jp) functional enrichment analyses. The correlation coefficient between samples was calculated to evaluate the repeatability among samples; the closer the

correlation coefficient is to 1, the better the consistency between two parallel experiments. DEGs related to specific immune-related functions were selected with a local script by searching the annotated keyword "immune".

The transcriptomic data are publicly available in the NCBI Short Read Archive (SRA) under Bioproject accession No.: PRJNA860310.

### Liquid chromatography with tandem mass spectrometry-based (LC-MS) non-targeted metabolomics detection and analysis

The Gene Denovo Biotechnology Co., Ltd. (Guangzhou, China) conducted the LC/MS-based non-targeted metabolomics experiments and data analysis according to the methods described by *Du et al. (2020)*. A total of 100 mg of each sample was first homogenized with one mL pre-cooling methanol-acetonitrile-water (2:2:1, v/v) solution three times (6.0 M/S, 20 s) by a Geno-grinder 2000 (SPEX) and spun down for 30 min (5, 40 kHz). After resting for 60 min ($-20\ °C$), each sample was centrifuged at 13,000 g for 15 min ($4\ °C$), and the supernatant was concentrated using a Termovap Sample Concentrator (DC-24; Shanghai, China). Then, the dry residue was redissolved by adding 100 µL acetonitrile-water (1:1, v/v), homogenized three times (6.0 M/S, 20 s), and centrifuged at 14,000 g for 15 min ($4\ °C$) to obtain the final supernatant. The chromatographic separation of metabolites was performed using a UHPLC system (Agilent Technologies, CA, USA), comprising an Agilent 1290 Infinity HILIC and an accurate-mass 6600 TOF-LC–MS (AB SCIEX Foster City, CA, USA) with a dual electrospray ionization (ESI) source in both positive (POS) and negative (NEG) ion mode. Baseline filtering, peak recognition, integration, retention time correction, and peak alignment were conducted for data preprocessing. After detecting the characteristic peak by mapping the MS and MS/MS mass spectrometry with the metabolic-specific database, the metabolites were identified, further normalized with Pareto scaling, and log-transformed.

### Bacterial 16S rRNA gene sequencing and analysis

Total genomic DNA from each biological sample was extracted using a Magen Hipure Stool DNA kit (Magen, Guangzhou, China) according to the manufacturer's instructions. The bacterial V3-V4 region of the 16S rRNA genes was amplified using the 341F (CCTACGGGNGGCWGCAG) and 806R (GGACTACHVGGGTATCTAAT) universal prokaryotic primers (*He et al., 2019a*) with the addition of an 8bp unique barcode and the required Illumina adapters. Samples were amplified in triplicate and replicate PCR reactions for each piece were pooled and purified with AMPure XP beads (Beckman Coulter, Brea CA, USA). A single composite sample for sequencing was prepared by combining approximately equal amounts of PCR products from each sample. Sequencing was performed on the Illumina Novaseq 6000 platform (in PE250 mode) at the Gene Denovo Biotechnology Co., Ltd. (Guangzhou, China).

FASTP was applied for raw data filtration. The paired-end clean data were then merged by FLASH v1.2.11 and analyzed according to the QIIME v1.9.1 pipeline. FASTP was also used for data quality control, and UCHIME was used for chimeric removal. The qualified sequences were then clustered into operational taxonomic units (OTUs) of $\geq 97\%$

similarity by UPARSE v9.2.64. The sequence with the highest abundance was selected as a representative sequence within each OTU cluster. Five microbial diversity indices were calculated: Goods_coverage, Chao1, phylogenetic distance (PD), Shannon, and Simpson indices. Four distance matrices were used to describe the dissimilarities among samples: Bray-Curtis, Jaccard, Weighted Unifrac, and Unweighted Unifrac distances based PCoA.

The bacterial 16S rRNA gene sequencing data are publicly available under Bioproject accession No.: PRJNA860311.

## Statistical analyses

The Wilcoxon test determined significant differences in the alpha indices of the groups. Significant differences in the microbial communities of the groups were determined using the *Anosim* and *Adnois* non-parametric multivariate statistical tests with a $P < 0.05$ significance threshold. The linear discriminant analysis effect size (LEfSe) was conducted to identify significant differences in the abundant taxa between the groups based on the Wilcoxon tests, with an LDA score of 4.0 and a $P$-value of 0.05. The Procrustes analysis was conducted using the Vegan R package to compare the congruence between: the transcriptomic profile and the microbiome, the transcriptomic profile and metabolism, and the microbiome and metabolism based on the goodness of fit (m2) $P$-value calculated by the Procrustes test and further verified by the Mantel test. The co-occurrence network was constructed using Cytoscape v3.9.0 and visualized using Gephi (https://github.com/gephi/gephi). Within the network, each connected edge represents a strong ($|r| > 0.8$) and significant ($P < 0.01$) Spearman's correlation (*Li et al., 2015*).

## RESULTS

### Expression pattern between cystic Peyer's patches (PPS) and the adjacent normal intestine tissues (NPPS)

To reveal the expression characteristics of the specific PPS of *Bactrian camels*, a high-throughput transcriptomic analysis of six samples from the cystic PPS and the adjacent NPPS of the cecum was performed. After data preprocessing and qualification, including data trimming, error removal, and quality control, 369,086,900 high-quality reads were obtained for reference genome alignment, and 330,364,407 were successfully mapped (average overall alignment rate $\approx$ 89.94%) and used for gene assembly. The correlation analysis among samples was evaluated by the Pearson correlation coefficient, suggesting that representatives from the same group showed consistency and biological repeatability (Fig. S1). The PCA plot showed that samples from PPS were separated from the NPPS group, while the PPS group was clustered more tightly (Fig. 1A). A Venn diagram was drawn to show the unique and overlapping genes between the two groups (Fig. 1B). A total of 9,257 genes were shared between PPS and NPPS, while 3,910 and 174 genes were specific to PPS and NPPS, respectively. Each 20 most highly-expressed genes in both PPS and NPPS genes are listed in Table 1, with five of these overlapping in the PPS and NPPS groups. Three housekeeping genes (*RPL0*, *RRPL10*, and *RPS2*), eukaryotic translation elongation factor 1 alpha 1 (*EEF1A1*), and the polymeric immunoglobulin receptor (*PIGR*), which transports IgA into mucosal secretions, all performed their regular

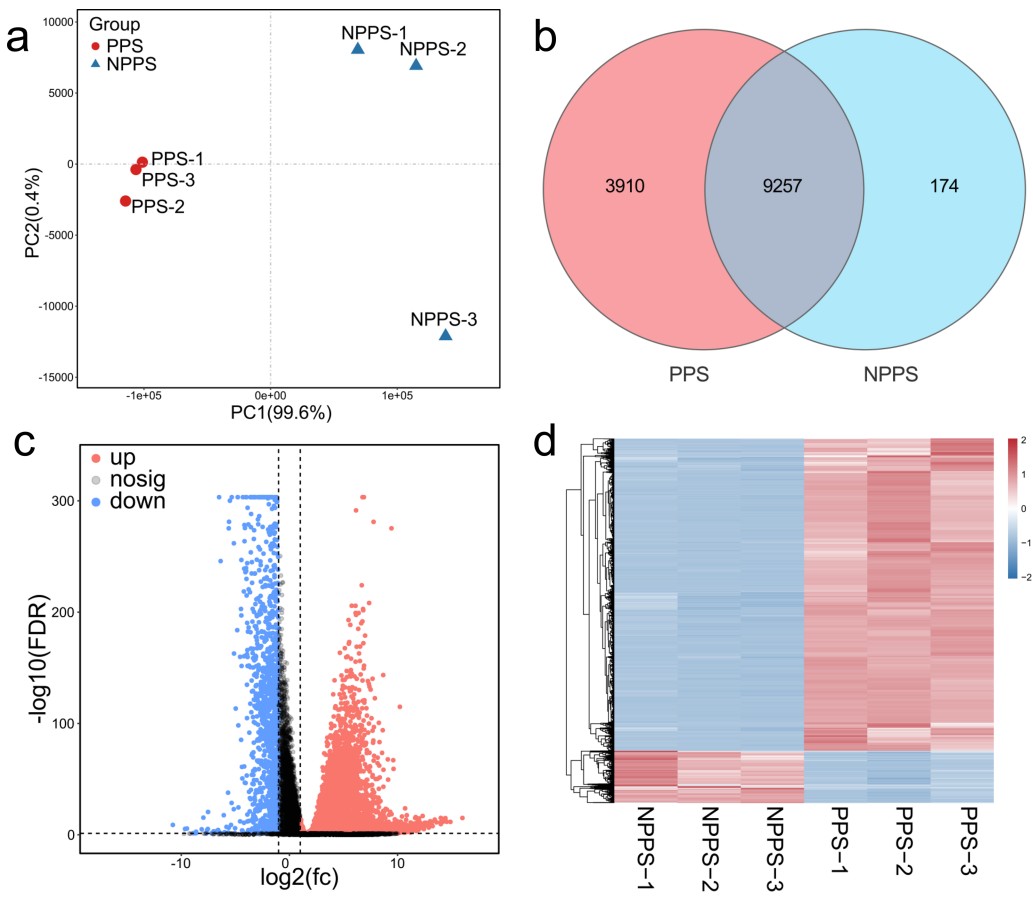

**Figure 1** **Comparison of gene expression profiles between the PPS and NPPS groups.** (A) The PCA shows the global gene expression level distinctions between the PPS and NPPS groups. The red circle represents samples from the PPS group, and the blue triangle represents samples from the NPPS group. (B) Venn diagram shows that 9,257 genes are overlapped by two groups, while 3,910 genes are unique to the PPS group and 174 genes are unique to the NPPS group. (C) The volcano plot shows the distribution of differentially expressed genes. The red dots represent the genes significantly up-regulated, while the blue dots represent the genes significantly down-regulated. The gray dots represent the genes with no significant differences between groups. (D) Heat map analysis shows the upregulated or down-regulated gene clusters originated from different samples.

functions in intestinal immune responses in the cecum of *Bactrian camels*. PPS had a higher expression of genes related to cellular immune functions like *JCHAIN*, immunoglobulin lambda-like polypeptide 5 (*IGLL5*), IgG1 heavy chain (*IGHG1*), Immunoglobulin heavy G2 chain (*IGHG2*), T cell lineage-specific genes related to the cluster of differentiation (*CD74*), and Beta2-microglobulin (*B2M*).

A total of 13,341 genes were selected for the analysis of differentially expressed genes (DEGs) based on a stringent FDR cutoff < 0.05 and log2 (FC) > 1. A total of 7,568 up-regulated DEGs and 1,266 down-regulated DEGs were obtained by comparing PPS to the NPPS group (Fig. 1C). A hierarchical clustering heatmap of all DEGs showed the strength of different clustering patterns between groups (Fig. 1D). Of the selected 8,834 DEGs, 8,081

**Table 1  Highly expressed genes in the NPPS and PPS groups.**

| Gene id | Symbol | Description | FPKM (mean ± sd, $n = 3$) |
|---|---|---|---|
| a. Highly expressed genes in NPPS | | | |
| ncbi_105064770 | EEF1A1 | Elongation factor 1-alpha 1 | 6883.46 ± 1117.13 |
| ncbi_105066579 | CA2 | Carbonic anhydrase 2 | 5074.88 ± 299.37 |
| ncbi_105076271 | KRT8 | Keratin, type II cytoskeletal 8 | 4357.19 ± 453.29 |
| ncbi_105075644 | IFITM1 | Interferon-induced transmembrane protein 1-like | 3692.05 ± 438.86 |
| ncbi_5309936 | MT-ND4L | NADH dehydrogenase subunit 4L | 2920.09 ± 1018.35 |
| ncbi_105065567 | ACTG1 | Hypothetical protein Y1Q_0007443 | 2518.12 ± 322.97 |
| ncbi_105080560 | RPLP0 | 60S acidic ribosomal protein P0 | 2461.30 ± 430.10 |
| ncbi_105074600 | LGALS3 | Galectin-3 | 2343.55 ± 113.83 |
| ncbi_105065683 | TSPAN1 | Tetraspanin-1 isoform X1 | 2315.68 ± 109.17 |
| ncbi_105076062 | RPL10 | 60S ribosomal protein L10 | 2312.69 ± 406.26 |
| ncbi_105075423 | PIGR | Polymeric immunoglobulin receptor | 2271.58 ± 221.32 |
| ncbi_105069768 | RPLP2 | 60S acidic ribosomal protein P2 isoform X1 | 2121.37 ± 206.99 |
| ncbi_105079514 | SLC25A6 | ADP/ATP translocase 3, partial | 2071.78 ± 366.85 |
| ncbi_105064863 | RPS2 | rCG34378, isoform CRA_i | 2055.23 ± 312.69 |
| ncbi_105084062 | UBB | Polyubiquitin-B isoform X3 | 2019.68 ± 276.77 |
| ncbi_105062636 | RPS11 | 40S ribosomal protein S11 | 1970.82 ± 185.05 |
| ncbi_105074446 | RPS5 | 40S ribosomal protein S5 isoform X1 | 1923.70 ± 333.84 |
| ncbi_105079327 | KRT19 | Keratin, type I cytoskeletal 19-like | 1860.59 ± 247.02 |
| ncbi_105065829 | CA4 | Carbonic anhydrase 4 | 1686.31 ± 63.05 |
| ncbi_105067478 | RPL32 | 60S ribosomal protein L32 | 1655.82 ± 279.68 |
| b. Highly expressed genes in PPS | | | |
| ncbi_105064770 | EEF1A1 | Elongation factor 1-alpha 1 | 12228.77 ± 263.54 |
| ncbi_105080402 | JCHAIN | Immunoglobulin J chain | 5640.98 ± 673.26 |
| ncbi_105077755 | TMSB4 | AAH16732.2 TMSB4X protein, partial | 3544.59 ± 22.53 |
| ncbi_105061888 | RPS6 | 40S ribosomal protein S6 | 3519.17 ± 114.93 |
| MSTRG.7185 | IGLL5 | Immunoglobulin lambda light chain precursor, partial | 3501.49 ± 334.60 |
| ncbi_105076977 | B2M | beta-2-microglobulin | 3299.10 ± 101.06 |
| ncbi_105066126 | TPT1 | Hypothetical protein FD755_015622 | 3161.59 ± 187.74 |
| MSTRG.14532 | IGHG1 | Ig gamma-3 chain C region | 3142.37 ± 791.50 |
| ncbi_105074299 | CD74 | HLA class II histocompatibility antigen gamma chain isoform X1 | 2867.34 ± 214.82 |
| ncbi_105079710 | RPS3A | 40S ribosomal protein S3a | 2676.10 ± 54.07 |
| ncbi_105066093 | RPL7 | 60S ribosomal protein L7 | 2650.52 ± 41.00 |
| ncbi_105073278 | RPL23 | 60S ribosomal protein L23 | 2619.79 ± 182.23 |
| ncbi_105064200 | RPS10 | 40S ribosomal protein S10 | 2429.74 ± 79.79 |
| ncbi_105075339 | RPS24 | 40S ribosomal protein S24 isoform e | 2401.06 ± 70.66 |
| ncbi_105064250 | HLA-DRA | Mamu class II histocompatibility antigen, DR alpha chain | 2387.51 ± 202.75 |
| ncbi_105076062 | RPL10 | 60S ribosomal protein L10 | 2317.63 ± 131.41 |
| MSTRG.14534 | IGHG2 | Ig gamma-3 chain C region, partial | 2316.47 ± 594.10 |
| ncbi_105080560 | RPLP0 | 60S acidic ribosomal protein P0 | 2274.07 ± 150.11 |

**Table 1** (*continued*)

| Gene id | Symbol | Description | FPKM (mean ± sd, $n = 3$) |
|---------|--------|-------------|----------|
| ncbi_105075423 | PIGR | Polymeric immunoglobulin receptor | 2262.23 ± 672.86 |
| ncbi_105064863 | RPS2 | rCG34378, isoform CRA_i | 2228.79 ± 88.34 |

**Notes.**
FPKM stands for Fragments Per Kilobase Million.

DEGs had annotations. The up-regulation of genes with a high fold-change was primarily related to immunomodulation, inflammation, and lipid homeostasis. For example, the CD5 antigen-like precursor (*CD5L*), V-set pre-B cell surrogate light chains (*VPREB1*, *VPREB3*), Fc receptor-like 1 (*FCRL1*), chemokine CXC ligand 13 protein (*CXCL13*), zymogen granule protein 16 homolog B (*ZG16B*), and probable G-protein coupled receptor 174 (*GPR174*) were all highly up-regulated in PPS. Conversely, the down-regulation of DEGs with a high fold-change was related to multiple functions, including transmembrane protein or aquaporin (*e.g.*, growth/differentiation factor 1 (*GDF1*), fatty acid-binding protein 6 gene (*Fabp6*), Aquaporin 8 (*AQP8*), transmembrane protein domain-containing 1 (*DCST1*), and transmembrane protein 182 (*TMEM182*); Table 2).

The enrichment analysis for gene ontology was performed to identify over-represented biological functions and classes from statistically significant DEGs. A GO functional enrichment analysis was categorized in terms of biological processes (BP), cellular components (CC), and molecular functions (MF). A total of 1,861 secondary classifications of BP, 192 secondary categories of CC, and 163 secondary types of MF (FDR < 0.05) were significantly enriched. Most DEGs were significantly enriched in the metabolic process (GO: 0008152, RF = 0.486), the nitrogen compound metabolic process (GO: 0006807, RF = 0.493), and the cellular metabolic process (GO: 0044237, RF = 0.489; Fig. 2A). Notably, positive regulation of the immune system process (GO: 0002684, RF = 0.648) and regulation of the immune system process (GO: 0002682, RF = 0.628) seemed to be most affected by group formation, indicated by a high RF value. A KEGG enrichment analysis was conducted to visualize specific functions, with 85 KO items significantly enriched (FDR < 0.05). According to the enrichment degree indicated by the rich factor, the enriched pathways mainly included mismatch repair, DNA replication, nucleotide excision repair, and intestinal immune network for IgA production (Fig. 2B). Moreover, the specific immune system-related pathways, including the detected intestinal immune network for IgA production (containing 66 up-regulated DEGs and three down-regulated DEGs), Th1/Th2/Th17 cell differentiation (containing 178 up-regulated DEGs and 12 down-regulated DEGs), the T cell receptor signaling pathway (containing 85 up-regulated DEGs and six down-regulated DEGs), and antigen processing and presentation (containing 70 up-regulated DEGs) were significantly enriched (Table S1).

The GO functional annotation identified 994 DEGs, including 890 up-regulated DEGs and 104 down-regulated DEGs, which participated in multiple immune-related functions (Table S2). A co-occurrence network analysis was performed to find the hub genes (most connected nodes within a network) in regulating the immunologic process (Table S3). The results showed that the differentially expressed genes, including Bruton tyrosine kinase

**Table 2  Top 20 differentially expressed genes in the NPPS and PPS groups.**

| Gene id | NPPS (mean, n = 3) | PPS (mean, n = 3) | log2 (FC) | FDR | Symbol | Description |
|---|---|---|---|---|---|---|
| a. Top 20 significantly up-regulated genes | | | | | | |
| ncbi_105069346 | 0.00 | 65.33 | 16.00 | 0.00 | VPREB3 | Pre-B lymphocyte protein 3 |
| ncbi_105067880 | 0.00 | 23.88 | 14.54 | 0.00 | HPGDS | Hematopoietic prostaglandin D synthase |
| ncbi_105077409 | 0.00 | 21.95 | 14.42 | 0.00 | FAM162B | Protein FAM162B, partial |
| ncbi_105065046 | 0.00 | 20.76 | 14.34 | 0.00 | UPK1A | Uroplakin-1a |
| ncbi_105064835 | 0.00 | 20.31 | 14.31 | 0.00 | ZG16B | Zymogen granule protein 16 homolog B |
| ncbi_105070503 | 0.00 | 18.89 | 14.21 | 0.00 | FCRL1 | Fc receptor-like protein 1 |
| MSTRG.7218 | 0.00 | 18.42 | 14.17 | 0.00 | VPREB1 | Immunoglobulin omega chain-like protein |
| ncbi_105062175 | 0.00 | 16.36 | 14.00 | 0.00 | AICDA | Single-stranded DNA cytosine deaminase |
| ncbi_105064162 | 0.00 | 16.34 | 14.00 | 0.00 | PI16 | Peptidase inhibitor 16 isoform X1 |
| ncbi_105082488 | 0.00 | 16.07 | 13.97 | 0.00 | CD5L | CD5 antigen-like |
| ncbi_105080423 | 0.00 | 12.41 | 13.60 | 0.00 | CXCL13 | C-X-C motif chemokine 13 |
| ncbi_105065182 | 0.00 | 11.08 | 13.44 | 0.00 | P2RX5 | Purinergic receptor P2X5 isoform A-like protein |
| ncbi_105074929 | 0.00 | 10.90 | 13.41 | 0.00 | SHISA3 | Protein shisa-3 homolog |
| ncbi_105075614 | 0.00 | 9.66 | 13.24 | 0.00 | TMEM273 | Putative uncharacterized protein C10orf128 homolog isoform X1 |
| ncbi_105065499 | 0.00 | 9.08 | 13.15 | 0.00 | GPR174 | Probable G-protein coupled receptor 174 |
| ncbi_105065255 | 0.00 | 8.98 | 13.13 | 0.00 | EBF1 | Transcription factor COE1 isoform X2 |
| ncbi_105062978 | 0.00 | 8.30 | 13.02 | 0.00 | ZCCHC12 | Zinc finger CCHC domain-containing protein 12 |
| ncbi_105082277 | 0.00 | 8.16 | 12.99 | 0.00 | ICOS | Inducible T-cell costimulator |
| ncbi_105075434 | 0.00 | 7.58 | 12.89 | 0.00 | FAM72A | Protein FAM72A isoform X1 |
| ncbi_105070211 | 0.00 | 7.45 | 12.86 | 0.00 | Clic6 | Chloride intracellular channel protein 6, partial |
| b. Top 20 significantly down-regulated genes | | | | | | |
| ncbi_105065706 | 0.80 | 0.00 | −9.65 | 0.00 | Gng13 | Guanine nucleotide-binding protein G(I)/G(S)/G(O) subunit gamma-13 isoform X1 |
| ncbi_105082589 | 0.71 | 0.00 | −9.48 | 0.00 | GDF1 | Embryonic growth/differentiation factor 1 |
| ncbi_105065237 | 0.40 | 0.00 | −8.63 | 0.00 | FABP6 | Gastrotropin |
| ncbi_105075917 | 0.28 | 0.00 | −8.11 | 0.00 | C10orf120 | Uncharacterized protein C10orf120 homolog isoform X1 |
| ncbi_105075380 | 3.31 | 0.01 | −7.96 | 0.00 | PLA2G12B | Group XIIB secretory phospholipase A2-like protein isoform X1 |
| ncbi_105062243 | 0.20 | 0.00 | −7.62 | 0.00 | GSG1 | Germ cell-specific gene 1 protein isoform X1 |
| ncbi_105074886 | 0.19 | 0.00 | −7.54 | 0.00 | CA7 | Carbonic anhydrase 7 |
| ncbi_105073757 | 483.70 | 2.63 | −7.52 | 0.00 | AQP8 | Aquaporin-8 |
| ncbi_105063539 | 0.15 | 0.00 | −7.20 | 0.04 | OTOS | Otospiralin |
| ncbi_105066926 | 0.13 | 0.00 | −7.02 | 0.00 | Olfr1020 | Olfactory receptor 1020-like |
| ncbi_105072225 | 0.12 | 0.00 | −6.91 | 0.00 | TMEM182 | Transmembrane protein 182 |
| ncbi_105068165 | 0.10 | 0.00 | −6.59 | 0.00 | DCST1 | DC-STAMP domain-containing protein 1 |
| ncbi_105070817 | 0.09 | 0.00 | −6.54 | 0.00 | KLHDC7B | Kelch domain-containing protein 7B |

**Table 2** (*continued*)

| Gene id | NPPS (mean, $n = 3$) | PPS (mean, $n = 3$) | log2 (FC) | FDR | Symbol | Description |
|---|---|---|---|---|---|---|
| ncbi_105072747 | 1437.36 | 15.84 | −6.50 | 0.00 | GUCA2B | Guanylate cyclase activator 2B |
| ncbi_105073399 | 74.80 | 0.91 | −6.36 | 0.00 | Cysrt1 | Cysteine-rich tail protein 1 |
| ncbi_105073400 | 0.08 | 0.00 | −6.32 | 0.03 | Fam166a | Protein FAM166A |
| ncbi_105070897 | 2.24 | 0.03 | −6.07 | 0.00 | Pla2g2c | Putative inactive group IIC secretory phospholipase A2 |
| ncbi_105064944 | 0.43 | 0.01 | −6.00 | 0.00 | ZXDB | Zinc finger X-linked protein ZXDB-like |
| ncbi_105072270 | 0.38 | 0.01 | −5.82 | 0.00 | NT5DC4 | 5′-nucleotidase domain-containing protein 4-like |
| ncbi_105082294 | 0.05 | 0.00 | −5.64 | 0.01 | MPP4 | MAGUK p55 subfamily member 4 |

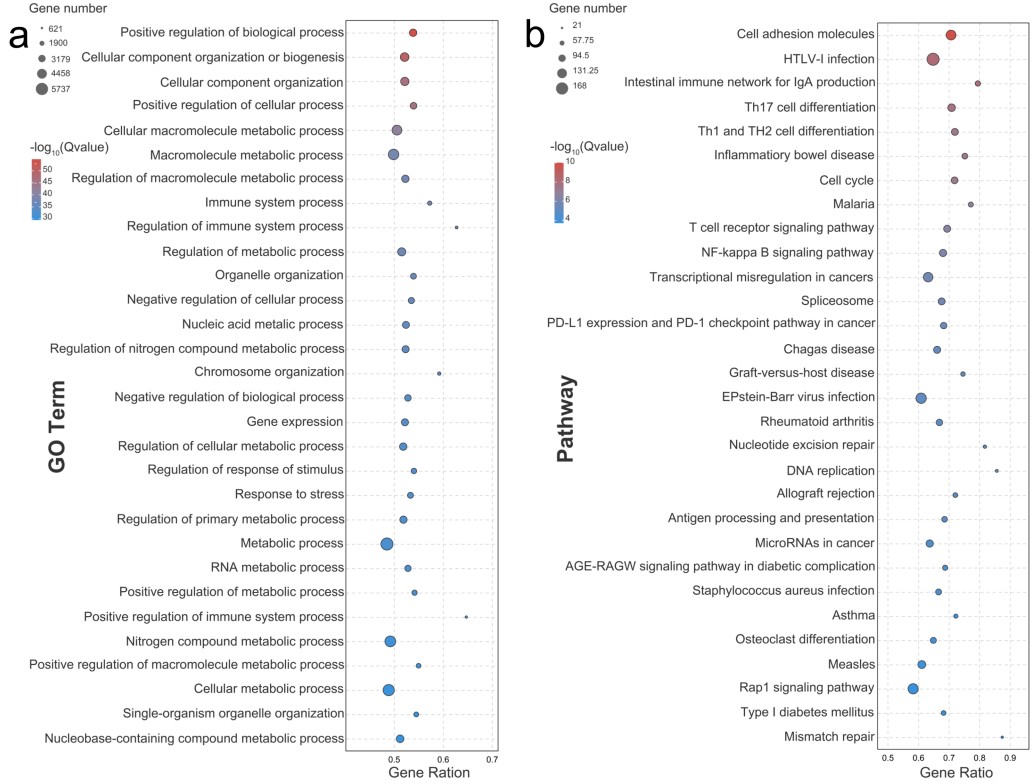

**Figure 2 Bubble diagram of the Top 30 GO terms and KEGG pathways enriched by differentially expressed genes.** Bubble diagram of the Top30 ranked GO terms (A) or Top30 ranked KEGG pathways (B) of the DEGs. The vertical axis indicates the specific terms, and the horizontal axis indicates the enrichment factor. The size of each dot indicates the number of genes in the GO term or KEGG pathway.

(*BTK*), P2X purinoceptor 7 (*P2X7R*), paired box protein (*Pax-5*), desmoglein-1 (*DSG1*), Tyrosine-protein phosphatase non-receptor type 12 (*PTPN12*), dedicator of cytokinesis protein 11 (*DOCK11*), T-box transcription factor protein 21 (*TBX21*), interleukin-10 (*IL10*), and HLA class II histocompatibility antigen DO beta chain (*HLA-DOB*), were hubs owning the biggest nodes' degrees (226) within the network.

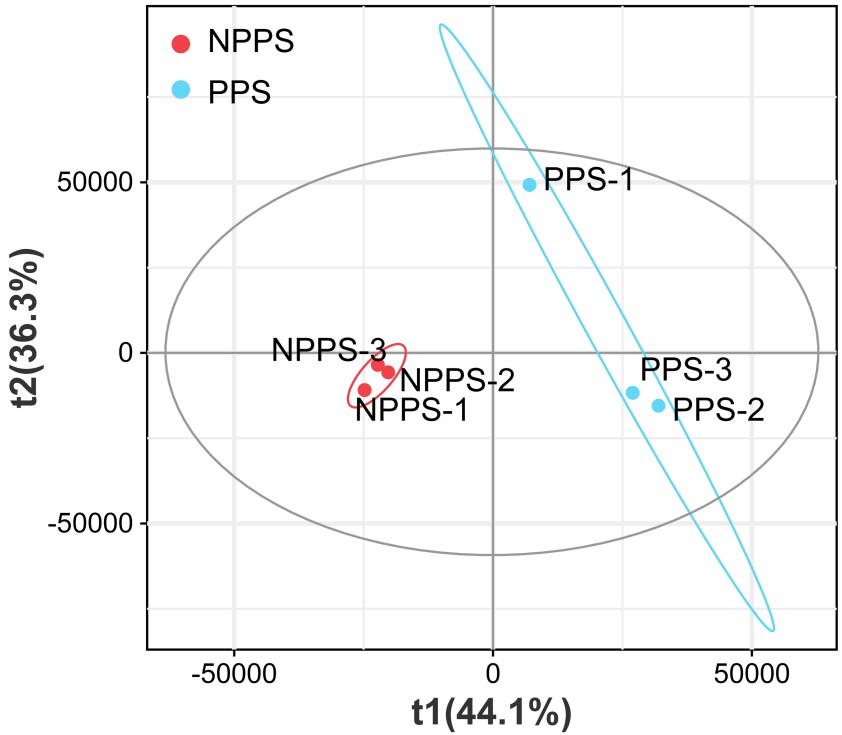

**Figure 3** Partial lease square-discriminant analysis (PLS-DA) analysis for visualization of groups' global metabolic profiles among NPPS and PPS samples.

## The composition of specific metabolites

To better understand the intestinal immunity of *Bactrian camels*, LC/MS-based non-targeted metabolomics were applied to assess the composition of the differential metabolites in the gut between Peyer's patches (PPS) and the adjacent normal intestine tissues (NPPS). A total of 18,514 metabolites were detected, including 11,394 positive ions and 7,120 negative ions, with 1,280 positive and 698 negative named ions identified. The partial least square-discriminant analysis (PLS-DA) was used for visualizing the global metabolic profiles of the groups. The PLS-DA found significant differences between the NPPS and PPS samples with 44.1% variation in PLS1 and 36.3% variation in PLS2 (Fig. 3); large variation was also observed within the PPS group. These results indicated that NPPS samples had similar metabolite compositions, but samples from the PPS group showed more individual differences.

To compare the significant differences in the composition of metabolites between PPS and NPPS samples, differentially expressed metabolites (DEM) were screened out in both ion-positive and ion-negative modes based on a VIP (variable important in the projection) $\geq 1$ in PLS-DA and a $P < 0.05$ in the $t$-test. Among the 1,978 total compounds, 113 DEMs were identified—12 up-regulated ($\sim$10.62%) and 101 down-regulated DEMs ($\sim$89.38%; Table S4). Compared with the results of the transcriptome, the DEMs were mostly down-regulated. The hierarchical cluster analysis (HCA) showed that the PPS group had a large number of compounds like Telmisartan, 2-piperidone, 1-Stearoyl-sn-glycerol

**Table 3  Differences in the four alpha diversity indexes between PPS and NPPS were examined by *Kruskal-Wallis*.**

| Alpha indices | PPS | NPPS | Kruskal-Wallis |
|---|---|---|---|
| Shannon | $5.13 \pm 0.51$ | $6.36 \pm 0.85$ | $P = 0.275$ |
| Simpson | $0.89 \pm 0.07$ | $0.92 \pm 0.06$ | $P = 0.375$ |
| Chao1 | $923.3 \pm 47.56$ | $1104.11 \pm 70.74$ | $P = 0.049$ |
| PD | $104.64 \pm 14.07$ | $105.9 \pm 6.69$ | $P = 0.827$ |

**Notes.**
Values represent the mean $\pm$ standard deviation ($n = 3$). $P < 0.05$ represented significant difference.

3-phosphocholine, and Deoxycarnitine, while the NPPS group accumulated significant-high 5-GLC tricin, Iselin, Agnuside, and Phenylbenzimidazolesulfonic acid. Deoxycholic acid (DCA; VIP = 37.96, log2FC = −2.97, $P = 0.00$), cholic acid (CA; VIP = 13.10, log2FC = −2.10, $P = 0.01$), and lithocholic acid (LCA; VIP = 12.94, log2FC = −1.63, $P = 0.01$) were the top contributors to the significant dissimilarities between the PPS and NPPS groups. These results indicated that the PPS and NPPS had similar overall metabolic profiles, but DEMs were mostly lower in PPS.

## Microbial communities varied among samples

A total of 592,024 qualified tags of six samples were obtained (98,671 per sample), and each sample was rarified to 93,469 tags for sample pre-normalization, followed by OTU clustering and community diversity analysis. The rarefaction curve of goods coverage, Chao1, and Shannon all reached saturation, indicating that the number of tags fit the microbial community complexity, which was enough for the subsequent analysis (Fig. S2). The Wilcoxon test found no significant difference in species diversity (Shannon and Simpson, all $P > 0.05$) between groups, while species richness (Chao1) was significantly lower ($P = 0.049$) in PPS ($923.30 \pm 47.56$) than in NPPS ($1104.11 \pm 70.74$; Table 3). A PCoA analysis based on four metrics (Bray-Curtis, Jaccard, weighted UniFrac, and unweighted UniFrac distances) revealed that samples from the different groups showed distinct patterns (Fig. S3). NPPS tended to cluster more tightly within the group, but samples from PPS were more dispersed, resulting in no significant differences between groups in *anosim* and *adnois* tests (all $P > 0.05$; Table S5).

Firmicutes ($35.92\% \pm 19.39\%$), Bacteroidetes ($31.73\% \pm 6.24\%$), and Proteobacteria ($13.96\% \pm 16.21\%$) were the three main phyla across all samples, but the relative abundance varied (Fig. S4A). At the genus level, *Lysinibacillus* ($11.19\% \pm 14.81\%$), *Escherichia/Shigella* ($8.48\% \pm 16.49\%$), *Bacteroides* ($7.22\% \pm 3.10\%$), *Synergistes* ($4.18\% \pm 5.15\%$), and *Rikenellaceae_RC9* ($3.53\% \pm 2.78\%$) had high abundance levels in the samples; samples from the same group showed similar compositions (Fig. S4B). The Venn diagram revealed that PPS and NPPS shared 19 phyla, while PPS had two unique phyla, Crenarchaeota and Chloroflexi (Fig. S5A). PPS and NPPS shared 75 families and 123 genera, while eight families and 23 genera were PPS group-specific, and nine families and 28 genera were NPPS group-specific (Figs. S5B and S5C); the unshared taxa all had low abundance levels in the samples.

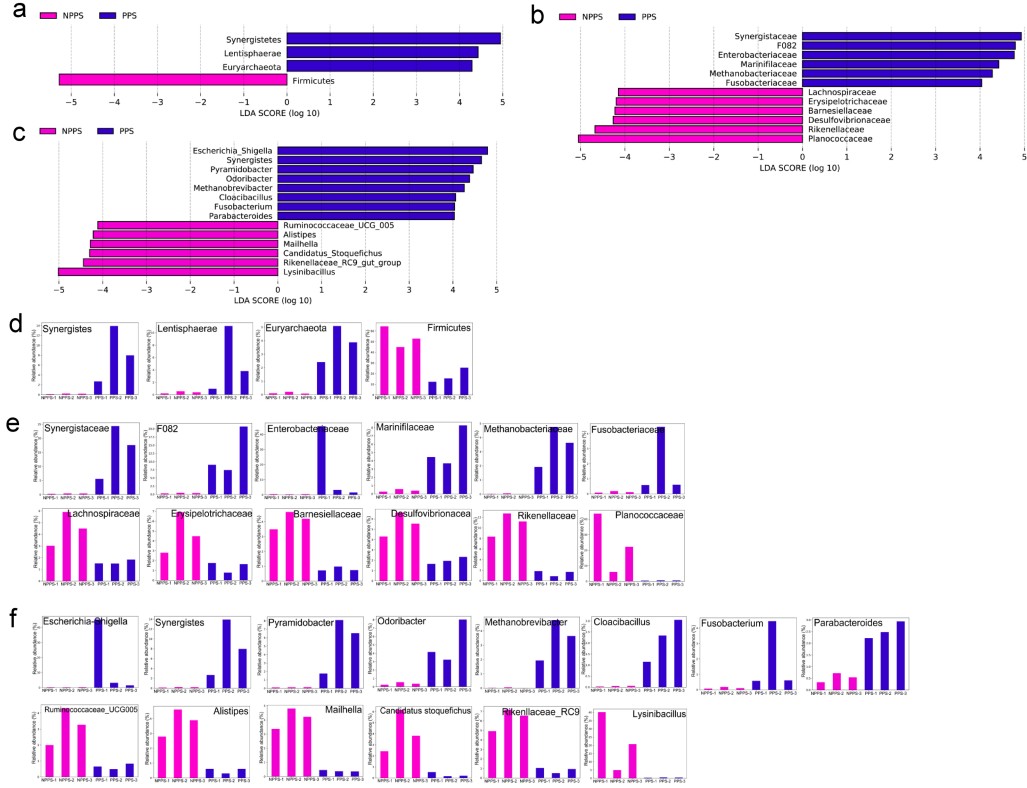

**Figure 4** **The linear discriminant analysis effect size (LEfSe) analysis of the differentially abundant taxa in PPS and NPPS groups.** Differentially abundant phyla (A), families (B), and genera (C) between NPPS (red) and PPS (green). The listed taxa are both statistically significant ($P < 0.05$) and had an LDA score ≥ 4.0. The bar chart shows the different abundance of biomarkers in each sample at the phylum level (D), at the family level (E), and at the genus level (F) corresponding to the identified taxa in (A) through (C).

A LEfSe analysis was conducted to identify the significantly different phyla (Fig. 4A), families (Fig. 4B), and genera between the different groups (Fig. 4C), with the identified indicator groups shown in a bar chart (Figs. 4D–4F). At the phylum level, the abundance levels of Synerdistetes, Lentisphaerae, and Euryarchaeota in PPS were significantly higher than in NPPS, while Firmicutes was considerably higher in the NPPS group. At the family level, Planococcaceae, Rikenellaceae, Desulfovibrionaceae, Barnesiellaceae, Erysipelotrichaceae, Lachnospiraceae, and Ruminococcaceae were significantly enriched in NPPS, while Synergistaceae, F082, Enterobacteriaceae, Marinifilaceae, Methanobacteriaceae, and Fusobacteriaceae were indicator features in the PPS group. At the genus level, *Lysinibacillus*, *Rikenellaceae_RC9_gut_group*, *Candidatus_Stoquefichus*, *Mailhella*, *Alistipes*, and *Ruminococcaceae_UCG_005* were biomarkers of the NPPS group, while *Escherichia/Shigella*, *Synergistes*, *Pyramidobacter*, *Odoribacter*, *Methanobrevibacter*, *Cloacibacillus*, *Fusobacterium*, and *Parabacteroides* were significantly higher in the PPS group.

**Relationships among expressed genes, metabolites, and microbial taxa**

A Procrustes analysis was used to find potential correlations among changes in the whole gene expression profiles, bacterial communities, and gut metabolites within all samples from different groups. The results showed that metabolic profiles were significantly correlated with bacterial communities ($P < 0.05$; Fig. 5A), indicating that changes in whole bacterial composition might alter the metabolic profile. However, gene expression had no significant associations with metabolites (Fig. 5B) or gut microbiota (Fig. 5C), indicating a change in gut metabolites or microbiota had no apparent effect on gene expression among samples during the steady stage. A Procrustes analysis was again used to find patterns in the changes among the profiles of DEGs, DEMs, and the differentially abundant taxa (DAT). DEGs still showed no significant correlation with DEMs (Fig. 5D) or DAT (Fig. 5E) ($P > 0.05$). In contrast, DEMs had significant associations with DAT using the Mantel test ($P = 0.01$) with the Procrustes test still showing no significance between DEMs and DAT ($P = 0.262$; Fig. 5F). Taken together, the transcriptome changes between groups showed no significant correlations with metabolites or microbial communities. In contrast, microbial community alterations were significantly correlated with the alteration of metabolites.

To better understand the highly-correlated associations between identified genera and DEMs, the co-occurrence network was constructed based on $|r| \geq 0.9$ and $P < 0.01$, and only interactions between genus and DEM were shown (Fig. 6), resulting in 120 negative and 208 positive associations within 69 taxa and 90 DEMs (41 positive ions and 49 negative ions). Within the network, seven DEMs (M403T65-neg, M329T119-neg, M309T38-neg, M277T42-2-neg, M473T27-neg, M747T38-1-pos, and M482t187-pos) and 14 genera (*Ruminococcaceae-UCG013*, *Peptococcus*, *Ruminococcus-1*, *Candidatus Stoquefichus*, *Prevotellaceae-UCG004*, *Ruminococcaceae-UCG009*, *dgA-11-gut-group*, *Sphaerochaeta*, *Ruminiclostridium-1*, *Erysipelatoclostridium*, *Akkermansia*, *Pygmaiobacter*, *Caproiciproducens*, and *Rikenellaceae-RC9-gut-group*) clustered much more tightly, producing 97 total associations, suggesting dense interactions.

## DISCUSSION

Peyer's patches (PPS) are secondary immune organoids in the intestinal mucosa and are mainly present along the intestine in nodular, faviform, cup-shaped, or cystic forms. Our previous study suggested that cystic PPS in the small intestine are unique to *Bactrian camels* (Qi et al., 2011) and further reported the distribution of cystic PPS in the cecum of *Bactrian camels* (ZhaXi et al., 2014). This study integrated transcriptome, untargeted metabolome, and 16S rDNA sequencing to expand the understanding of the immunological microenvironment of cystic PPS in the cecum of healthy *Bactrian camels*.

The mucosal immune cells (*e.g.*, T cells or B cells) frequently located in the surface epithelium along the intestine are immune active (Brandtzaeg, 2009). The immune-related *PIGR* gene was found to be highly co-expressed by both the PPS and NPPS groups (Table 1); *PIGR* was shown to have the dual role of transporting locally-produced dimeric IgA across mucosal epithelia and serving as the precursor of secretory component a glycoprotein that enhances the immune functions of SIgA (Johansen & Kaetzel, 2011). The annotated DEGs

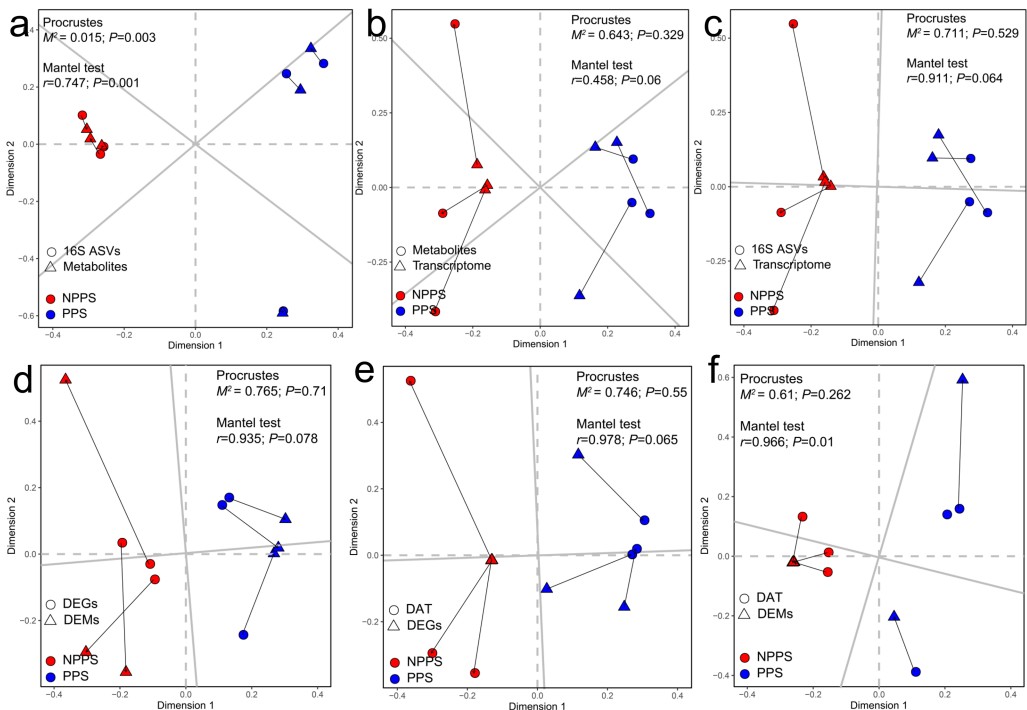

**Figure 5** **Relationships among the transcriptomic profiles, microbial community structure, and metabolic profiles by Procrustes analysis and Mantel test.** Relationship between the whole metabolic profiles and the microbial community structures (A), the relationship between the metabolic profiles and the transcriptomic profiles (B), the relationship between the transcriptomic profiles and the microbial community structures (C), the relationship between the differentially expressed gene (DEGs) and the differentially expressed metabolites (DEMs) (D), the relationship between the DEGs and the differentially abundant taxa (DAT) (E), and the relationship between DAMs and the DATs (F). The correlation is considered significant when $P < 0.05$.

with a high fold-change that were up-regulated in the PPS group were primarily related to immune regulation and lipid homeostasis (Table 2). For example, the *CD5L* gene has been reported as a key protein in regulating the immune homeostasis and inflammation (*Alonso-Hearn et al., 2019*). The *VPREB1* and *VPREB3* genes correlate with expressing B-cell progenitor populations (*Giladi et al., 2018*). The *FCRL1* gene has been shown to promote activation and function of B cell (*Zhao et al., 2019*). The *GPR174* gene participates in the calcium signaling of T cells (*Trebak & Kinet, 2019*), and the *ZG16B* gene regulates the Wnt/$\beta$-catenin pathway and enhances immunosuppressive activity (*Escudero-Paniagua et al., 2020*). The NPPS group had significantly higher expressions of genes related to multiple functions, including aquaporin (*e.g.*, *AQP8* (*Lv et al., 2022*)) or regulating cell growth and differentiation (*e.g.*, *GDF1 Rankin et al., 2000*).

The systematic functional analysis of DEGs reflected the functional specialization of the cystic PPS, and a GO enrichment analysis detected their potential functions. Nine hub DEGs (*BTK, P2RX7, Pax5, DSG1, PTPN2, DOCK11, TBX21, IL10,* and *HLA-DOB*) were all up-regulated and showed significant interactions between 994 immune-related DEGs (Table S3). Recent research suggested that *BTK* could support normal intestinal IgA

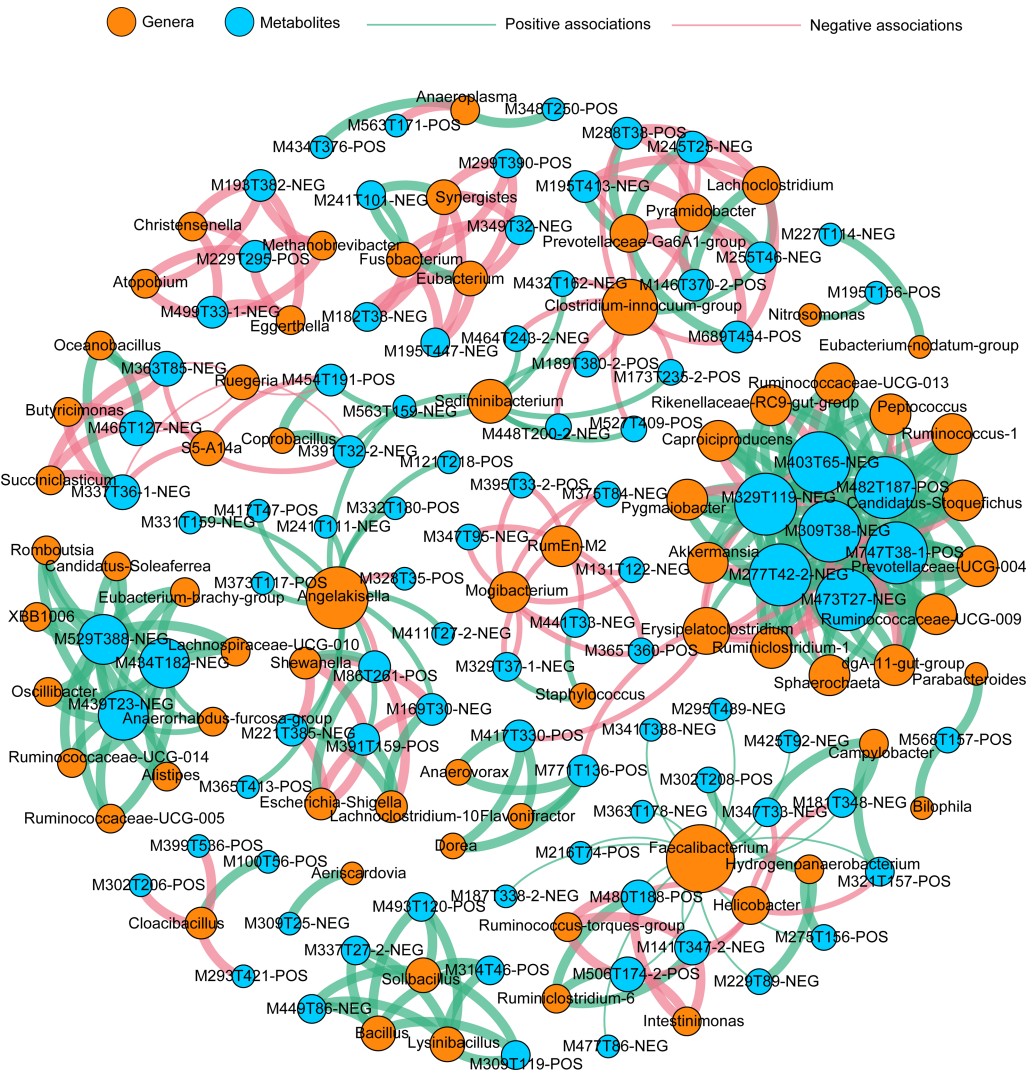

**Figure 6** **Co-occurrence network summarizing the highly correlated associations between identified genera and differentially expressed metabolites (DEMs).** A connection indicates a strong ($|r| > 0.90$) and significant ($P < 0.01$) Spearman's correlations and only interactions between genus and DEMs are shown. Red lines indicate negative correlations, while green lines indicate positive correlations. The size of each node is proportional to the number of connections: The thickness of each edge is proportional to the value of the Spearman correlation.

development to maintain gut mucosal immunity in response to commensals (*Bonami et al., 2022*). *Pax5* encodes the B-cell lineage-specific activator protein, which affects B-cell differentiation at the early stages (*Arseneau et al., 2009*). *DOCK11* regulates B-cell activation during immunization (*Sakamoto & Maruyama, 2020*). *TBX21* is known as a regulator of immune cell development and function (*Stolarczyk, Lord & Howard, 2014*), while *IL-10* has been shown to modulate inflammatory responses and is particularly important in maintaining intestinal microbe-immune homeostasis (*Neumann, Scheffold & Rutz, 2019*). Therefore, the nine hubs identified in this study, including the genes (*e.g., P2RX7, DSG1,*

and *PTPN2*) with uncertain roles, might be essential in regulating the PPS immune microenvironment in the cecum of *Bactrian camels*. A KEGG enrichment analysis further found that the immune system-related pathways were enriched mainly by the significant up-regulated DEGs, suggesting the immunological pathways actively take place in PPS.

The intestine is the main locus of direct interaction between the immune system and the vast number of microorganisms that play a pivotal role in guiding the maturation of the mucosal immune system and shaping systemic immunity. The gut contains metabolites produced and synthesized by the gut microbiota and host cells, which affect the stability of the intestinal microenvironment (*Rooks & Garrett, 2016*). The current omics revolution offers an unprecedented opportunity to explore the detailed composition and variations between different tissues. As expected, there were significant differences between the PPS and NPPS groups in this study, though higher heterogeneity was found in the PPS group (Fig. 3). Approximately 88% of DEMs were downregulated in PPS, with DCA, CA, and LCA identified as the three top contributors to group differentiation (Table S4). Previous studies suggest that only 1–2% of bile acids can be delivered from the liver to the intestine and further transformed for host utilization (*Sagar et al., 2015*). CA is a primary bile acid, while DCA and LCA are microbial-modified secondary bile acids that are involved in lipid absorption in the host intestine (*Gérard, 2013*). PA was significantly higher in PPS; a previous study confirmed that a high concentration of PA could stimulate intestinal IgA antibody production through direct or indirect pathways (*Kunisawa et al., 2014*).

The microbial communities of all samples were examined using 16S rDNA high-throughput sequencing. PPS samples could be differentiated from NPPS samples, but PPS samples showed a higher dispersion within the group (Fig. S3). These results were the same as the results from the untargeted metabolism analysis. The intestinal immune system often nourishes prosperous bacterial communities and establishes advanced symbiotic relationships (*Sutherland & Fagarasan, 2012*). Firmicutes, Bacteroidetes, and Proteobacteria were the three main phyla across all the samples in this study, which is consistent with our previous study that found that these phyla dominantly colonized on the surface of the different mucosal immune inductive sites of the gastrointestinal tract in *Bactrian camels* (*Zhang et al., 2020*). Proteobacteria is conducive to the homeostasis of the anaerobic environment of the GI tract gut (*Moon et al., 2018*), while the ratio of Firmicutes and Bacteroidetes impacts gut dysbiosis (*Grigor'eva, 2020*). In this study, the PPS group had a higher abundance of *Escherichia_Shigella*, *Synergistes*, *Pyramidobacter*, *Odoribacter*, *Methanobrevibacter*, *Cloacibacillus*, *Fusobacterium*, and *Parabacteroides* compared with the NPPS group. Previous studies have found that *Escherichia_Shigella* and *Fusobacterium* have virulence factors and might trigger pro-inflammatory activities to promote intestinal inflammation (*Dai et al., 2021*; *Kostic et al., 2013*). *Synergistes* and *Pyramidobacter* belong to the Synergistetes phylum and have been reported to degrade toxic matter such as pyridinediols or fluoroacetate (*Allison et al., 1992*; *Davis et al., 2012*). A high abundance of the butyrate-producing genus *Odoribacter* has been suggested to be health-promoting (*Fei & Zhao, 2013*), but no significant difference in butyrate-related metabolites was found between groups in this study, which might due to the relatively low abundance of the biomarkers. The methanogenic Euryarchaeote, *Methanobrevibacter*, a group of

acetogenesis and sulfate-reducing anaerobes, is pro-inflammatory (*Bang et al., 2014*) and has been shown to regulate the expression of adhesion-like proteins in proximity to the gut-associated lymphoid tissue (*Samuel et al., 2007*). The mucin-degrading genus *Cloacibacillus* might be important in regulating the gut environment (*Mu et al., 2019*), and the genus *Parabacteroides* can produce antagonistic substances to defend against the invasion of exogenous microorganisms (*Nakano et al., 2006*). The results of this study indicated that the microbiota in PPS were not all beneficial to the immune response of the gut, but the synergistic effects of these biomarkers might form a stable microenvironment in PPS. Host gut immunity helps maintain gut-microbe symbiosis and suppresses gut dysbiosis. In this study, PPS exhibited a significantly lower species richness, reflected by Chao1, compared with NPPS (Table 3), indicating that the immune microenvironment in PPS might have a stricter selective pressure on symbiotic flora than the normal fragments of the intestine.

A correlation analysis among all samples was conducted to identify the interactions among the expressed genes, gut metabolites, and gut microbiota. The results showed that the alteration of gene expression profiles had no significant correlation with gut metabolites or symbiotic taxa composition, while gut microbiota had significant correlations with gut metabolites (Fig. 5), indicating that the metabolites of intestinal tissues in the present study might mainly derive from the gut taxa. The different gene profiles had no apparent correlations with varying profiles of gut metabolites or gut taxa among PPS samples or the adjacent intestinal tissues. A co-occurrence network analysis found a small group of dense connections containing seven gut metabolites and 14 gut genera, which produced significant correlations among all of the samples in healthy *Bactrian camels* (Fig. 6). Among the 14 gut genera identified, most were reported to be host friendly. For instance, *Akkermansia* was an important genus in this study; *Akkermansia* can conduct mucin degradation in the intestine and has been suggested as a biomarker for a healthy intestine (*Derrien, Belzer & Vos, 2017*). *Yang et al. (2021)* found that the up-regulation of Candidatus *Stoquefichus* and *Akkermansia* might inhibit the production of inflammatory cytokines. *Caproiciproducens* is an acid-producing bacterium, which could improve host immunity by producing more caproic acid (*Zhang et al., 2019*). Whereas, the genera *Peptococcus* and *Erysipelatoclostridium* were reported as pathogens (*Cai et al., 2019*; *Cai et al., 2022*). In the study network, the genus *Erysipelatoclostridium* mainly showed negative correlations with the metabolites within the module. This study found that in healthy *Bactrian camels*, probiotics and pathogens co-exist and have positive or negative effects on gut metabolites.

## CONCLUSIONS

This study used multiple omics strategies to identify the immunological characteristics of species-unique Peyer's patches (PPS) in the cecum of healthy *Bactrian camels*. Samples from PPS were distinguishable from the NPPS group based on gene expression data, untargeted metabolomics data, and microbial community compositions. Changes in expressed genes had no significant associations with gut metabolites or microbial community changes among different samples, whereas alteration of gut taxa directly

led to changes in metabolites. As expected, the PPS group expressed more immune-related genes and an enrichment analysis revealed that nine hubs (*e.g.*, *BTK*, *P2RX7*, and *Pax5*) were key genes participating in multiple immunologic processes. Further, an untargeted metabolism analysis indicated that PPS and NPPS had a similar composition of most compounds, but 88% of DEMs were decreased in PPS, with CA, DCA, and LCA primary or secondary bile acids identified as the top contributors to group differentiation. PPS had a significantly lower species richness and the biomarker taxa contained both probiotics and pathogens, which might have a synergistic relationship that underlies the immunological microenvironment in cystic PPS. This research provides new insights into the understanding of cystic PPS in the cecum of *Bactrian camels*.

### Funding

This work was supported by the National Natural Science Foundation of China (Grant Nos. 31960693; 31760723). The funders had no role in study design, data collection and analysis, decision to publish, or preparation of the manuscript.

### Grant Disclosures

The following grant information was disclosed by the authors:
National Natural Science Foundation of China: 31960693, 31760723.

### Competing Interests

The authors declare there are no competing interests.

### Author Contributions

- Xiao shan Wang conceived and designed the experiments, analyzed the data, prepared figures and/or tables, authored or reviewed drafts of the article, and approved the final draft.
- Pei xuan Li performed the experiments, prepared figures and/or tables, and approved the final draft.
- Bao shan Wang performed the experiments, authored or reviewed drafts of the article, and approved the final draft.
- Wang dong Zhang analyzed the data, authored or reviewed drafts of the article, and approved the final draft.
- Wen hui Wang conceived and designed the experiments, authored or reviewed drafts of the article, and approved the final draft.

### Animal Ethics

The following information was supplied relating to ethical approvals (*i.e.*, approving body and any reference numbers):

The Animal Care and Use Committee (IACUC) of the College of Veterinary Medicine of Gansu Agricultural University approved all experimental procedures (Approval No: GSAU-AEW-2020-0010).
## DNA Deposition

The following information was supplied regarding the deposition of DNA sequences:

The transcriptomic data are available in the NCBI Short Read Archive (SRA): PRJNA860310. The bacterial 16S rRNA gene sequencing data are available at the SRA: PRJNA860311. The data of untargeted metabolism is available in the Supplemental File.

## Data Availability

The data is available at NCBI: PRJNA860310 and PRJNA860311. The data of untargeted metabolism is available in the Supplemental File.

## Supplemental Information

Supplemental information for this article can be found online at http://dx.doi.org/10.7717/peerj.14647#supplemental-information.

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
