# Peer review of "Integrated omics analysis reveals the immunologic characteristics of cystic Peyer’s patches in the cecum of Bactrian camels"

_PeerJ, doi:10.7717/peerj.14647_

## Round 0.1 · original submission · Major Revisions

The reviewers were generally positive, and I invite you to resubmit a revised manuscript where the suggestions of the reviewers are addressed.

Reviewer 1 ·

Basic reporting

The performed work is interesting with a clear rationale. A sufficient number of references has been used to support the ideas of the work. The English language of the manuscript needs to be improved.

Experimental design

The research is original. The methodology is solid and relevant to the aims of the work.

Validity of the findings

The findings of this work improve knowledge related to factors governing the functional adaptations of camel secondary immune organs to desert life. Proper statistical tests were used to analyze differences between experimental groups. The conclusions justify the aims of the work.

Additional comments

Dear authors,
I have read your paper carefully. The paper is interesting and advance our understanding of the immunologic signature of the Peyer`s patches in camel. Overall, the performed work is interesting. However, the findings of the omics studies lack both technical (using the analyzed specimens) and biological (utilizing other independent specimens from other camels) validations. A number of the differentially expressed genes should be confirmed/validated on individual basis using a more precise technique such as qPCR, western blotting, IHC, etc. Including such work will increase both the credibility and reliability of the work.
The language of the manuscript is not adequate and needs to be thoroughly revised. Several examples are listed in the minor comments below.
The use of technical terms should be checked. When an abbreviation to be used, please list it in full at first mention. Also, do not start paragraphs with numerical values or abbreviations.
In the abstract the abbreviations related to sampling is confusing for the reader. species-unique cystic Peyer's patches (PPs); cystic PPs samples (PPS); normal intestine tissues (NPPS). Also latter at line 122-123: cystic PPs and the adjacent no-PPs containing segments & lines 144-145: contents of PPs and the mucus of the non-PPs were collected. Using too many abbreviations for the same thing is not recommended and distract the readers. I advise to set only one abbreviation for Peyer’s patches and another one for the non-Peyer’s patches.
Minor comments:
Line 20: three youth Camelus bactrianus change to three young Bactrian camels.
Line 29: only 113 differentially abundant metabolites change to only 113 differentially expressed metabolites.
Line 59: Camelus bactrianus have evolved change to Bactrian camels have evolved.
Line 62: for functional maintenance of what, the individual?
Line 71: Previous studies reported that distributed along the whole small intestine, please fix!
Line 80: transcriptome to research the regional heterogeneity in rat or pigís PPs change to PPs of rat and pig.
Line 136: Three clinically normal Bactrian camels (4 to 5 years old, two females and one male), why didn’t you unify the sex?
Line 137: All the camels were brought from the slaughterhouse, do you mean camels or camel specimens?
Line 181: LC/MS-based non-targeted metabolomics detection and analysis, what does LC/MS stand for? Write in full, then abbreviate.
Line 216: Then, Goods_coverage, Chao1, phylogenetic distance (PD), Shannon, and Simpson indices were calculated. These are microbial diversity metrics, please add a hint here about this.
Line 235: Sequence accessions: move contents of this section to the end of relevant sections (Transcriptomic sequencing and analyzing and Bacterial 16S rRNA gene sequencing and analysis respectively).
Line 241: Expression pattern between cystic PPs and non-PPs region, Write in full, then abbreviate.
Line 310: To better comprehend the intestinal immunity of Camelus bactrianus under different microbial
communities, we applied the LC/MS-based non-targeted metabolomics to assess the differential
guts' metabolites compositions between the PPs and non-PPs regions. Too lengthy and confusing.
Line 396: 14 genera (e.g. Ruminococcus-1, Stoquefichus, Ruminococcaceae-UCG-009, and Akkermansia), please list all genera.
Line 407: at the regular stage of Camelus bactrianus, what do you mean by regular?
Line 496 : gene expression profiles had no significant correlation ship with the gut metabolites. Do you mean correlation?
Line 518: We first time set out to determine the immunological microenvironment of cystic PPs in the
cecum of Camelus bactrianus at the normal living stage by multiple omics strategies. Please fix!
Line 563: reference Cai et al. lacks volume number and article ID. The same applies to Sagar et al., Wang et al., Yang et al.
Table 1. Highly expressed genes of the NPPS or PPS group. Change to “Highly expressed genes in the NPPS and PPS groups”
Add a table footnote descripting abbreviations used in the table e.g., FPKM.
Table 2. TOP20 up-regulated and down-regulated genes ranked by log2 (FC). Change to “Top 20 differentially expressed genes in the NPPS and PPS groups”. Inside the table, adjust the letter case of TOP20.
Table 3. Significant differences in the four alpha indexes between the two groups were examined by Kruskal-Wallis. Change to “Differences in the four alpha diversity indexes between PPS and NPPS were examined by Kruskal-Wallis”. In the footer: P < 0.05 represented difference significant? significant difference!
Figure 1 legend: change the word “showed” to “showing”.
Figure 2 legend: The bubble diagram of the TOP30 GO terms or KEGG pathways enriched by DEGs. Change to “Bubble diagram of the Top 30 GO terms and KEGG pathways enriched by differentially expressed genes”.
Figure 3 legend: delete “The”.
Figure 4 legend: delete “The results showed the”.
Figure 5 legend: delete “The results showed the”. Delete “DAT represented the diferentially abundant taxa”, then fix the text.
Figure 6 legend: change “were shown” to are shown. Delete “DAMs represented the diferentially abundant metabolites”, then fix the text.

·

Basic reporting

Please unify the typing of Camelus bactrianus in italic form throughout the manuscript

Experimental design

No comment

Validity of the findings

No comment

Reviewer 3 ·

Basic reporting

no comment

Experimental design

no comment

Validity of the findings

no comment

Additional comments

By using a variety of omics techniques, the authors have studied the immunological microenvironment of cystic PPs in the cecum of Camelus bactrianus at the normal living stage. The results are presented in an orderly fashion, and the introduction is well written. There are no significant issues. I recommend the publishing following a minor adjustment.
The present tense should be used in the title of the article.
The authors could make the figures more clear

---

## Round 0.2 · accepted · Accept

Thanks you for the revisions, which was well received by the reviewers.

Reviewer 1 ·

Basic reporting

The paper improved from the initial submission.

Experimental design

The experimental design matches the rationale and achieves the obtained data.

Validity of the findings

The data are valid.

Additional comments

Most of my comments have been addressed.